Transcriptome profiling reveals stress-responsive gene networks in cattle muscles

Cassar-Malek Isabelle 1 isabelle.cassar-malek@inrae.fr
Pomiès Lise 1 2
de la Foye Anne 1
Tournayre Jérémy 1
Boby Céline 1
Hocquette Jean-François 1
1 Université Clermont Auvergne, INRAE, VetAgro Sup, UMR Herbivores , Saint-Genès-Champanelle , France
2 Université de Toulouse, INRAE, UR MIAT , Castanet-Tolosan , France
Li Cong-Jun
Electronic publication date: 2022 Apr 6
Publication date: 2022
Volume: 10
Electronic Location ID: e13150
Received 2021 Dec 16; Accepted 2022 Mar 1
Copyright: © 2022 Cassar-Malek et al.
Copyright year: 2022
Copyright holder: Cassar-Malek et al.
License: This is an open access article distributed under the terms of the Creative Commons Attribution License, which permits unrestricted use, distribution, reproduction and adaptation in any medium and for any purpose provided that it is properly attributed. For attribution, the original author(s), title, publication source (PeerJ) and either DOI or URL of the article must be cited.
License URL: https://creativecommons.org/licenses/by/4.0/

Keywords: Pre slaugther stress, Cattle, Transcriptomics, Muscle response, Transcription factors

Funding: French National Agency for Research 6th Framework Programme of the European Union FOOD-CT-2006-36241 This work was supported by the “Lipivimus” Research program (2007–2009) funded by the French National Agency for Research and the Prosafe-Beef Integrated Project (Contract no. FOOD-CT-2006-36241) supported under the 6th Framework Programme of the European Union. The funders had no role in study design, data collection and analysis, decision to publish, or preparation of the manuscript.

==============================
In meat-producing animals, preslaughter operations (e.g., transportation, mixing unfamiliar animals, food and water deprivation) may be a source of stress with detrimental effects on meat quality. The objective of this work was to study the effect of emotional and physical stress by comparing the transcriptomes of two muscles (M. longissimus thoracis, LT and M. semitendinosus, ST) in Normand cows exposed to stress (n = 16) vs. cows handled with limited stress (n = 16). Using a microarray, we showed that exposure to stress resulted in differentially expressed genes (DEGs) in both muscles (62 DEGs in LT and 32 DEGs in ST, of which eight were common transcription factors (TFs)). Promoter analysis of the DEGs showed that 25 cis transcriptional modules were overrepresented, of which nine were detected in both muscles. Molecular interaction networks of the DEGs targeted by the most represented cis modules helped identify common regulators and common targets involved in the response to stress. They provided elements showing that the transcriptional response to stress is likely to (i) be controlled by regulators of energy metabolism, factors involved in the response to hypoxia, and inflammatory cytokines; and (ii) initiate metabolic processes, angiogenesis, corticosteroid response, immune system processes, and satellite cell activation/quiescence. The results of this study demonstrate that exposure to stress induced a core response to stress in both muscles, including changes in the expression of TFs. These factors could relay the physiological adaptive response of cattle muscles to cope with emotional and physical stress. The study provides information to further understand the consequences of these molecular processes on meat quality and find strategies to attenuate them.

Introduction

In livestock species, psychological/emotional factors (including unfamiliar environment or social mixing), physical factors (including transportation, noise or vibrations), climatic factors (including temperature and humidity associated with transfer to the slaughterhouse), and deprivation of food and water are sources of emotional and physical stress. The exposure of animals to stressful conditions has several adverse impacts, including metabolic and health problems and poor welfare. Most of the above conditions often occur during farm management, during preslaughter operations (Deters & Hansen, 2020) or during slaughter, and they have detrimental effects on nutritional and organoleptic meat quality, as reported in pigs and poultry (Debut et al., 2005; Debut et al., 2003; Monin & Sellier, 1985). There is also compelling evidence that preslaughter stress has an undesirable effect on meat quality traits (e.g., low tenderness and juiciness) in both beef and lamb (Ferguson & Warner, 2008; Muchenje et al., 2009; Terlouw, 2015; Terlouw et al., 2008; Terlouw et al., 2021; Reiche et al., 2019). The impact of stress on meat quality has been explained by changes in physiological and metabolic properties of the muscle, which is converted to meat postmortem: higher depletion of glycogen before slaughter, less production of lactic acid (a byproduct of post-mortem glycolysis), and thus insufficient pH decline. Muscles with borderline pH (5.9–6.1) end up being very tough (Marsh et al., 1987), leading to a defect known as dark cutting beef or dark, firm, and dry (DFD) meat.

Changes may be related to variations in protein activities, as well as in protein levels driven by changes in gene expression. Some studies have reported alterations in the muscle proteome of farmed pigs (Morzel et al., 2004), chickens (Hazard et al., 2011; Zanetti et al., 2013), fishes (Silva et al., 2012) and cattle (Díaz et al., 2020; Sentandreu et al., 2021). However, there are few data on the transcriptional response to stress in the muscle of meat-producing animals despite few studies in pigs (Davoli et al., 2009) and in steers following surgery (Zhao et al., 2012). Herein, we examined the transcriptomic response in two different skeletal muscles from cows exposed to preslaughter stress conditions. We used these responses to infer stress-induced changes in the biological and physiological function of these muscles and discussed the biological functions affected by exposure to stress of psychological and physical origins and their potential impact on meat quality.

Materials and Methods

Animals and samples

The experiment was conducted with 32 pure Normand cull cows of 48–60 months of age purchased from different private farms in western France. The cows were not pregnant or lactating and had a medium fatness score. They were housed in the experimental farm of the INRAE research centre (UE Herbipôle-Low Mountain Ruminant Farming Systems Facility; DOI 10.15454/1.5572318050509348E12) as described by Gobert et al. (2009) and Delosière et al. (2020).

The cows (mean live weight of 642 kg) received a straw (30%) and concentrate (70%) based diet supplemented with lipids (40 g oil/kg diet DM) obtained from extruded linseeds (60%) and rapeseeds (40%) for 101 ± 3 d. For one group of cows, this diet was the control diet described in Delosière et al. (2020). For the other group, the diet was supplemented with vitamin E (155 IU/kg diet DM) and plant extracts rich in polyphenols (INRA patent #P170-B-23.495 FR; c7 g/kg diet DM, respectively; EP diet). Supplementation with vitamin E (lipophilic antioxidant) and plant extracts rich in polyphenols was used to protect against beef lipoperoxidation in a previous study on sheep (Gladine et al., 2007). The cows received a morning and evening meal representing a daily quantity of 1.8 kg of concentrate and 0.8 kg of hay. They had free access to water. The cows received an isoenergetic and isonitrogenous ration for a mean daily body weight gain of 1.6 kg over the 101 ± 3 d finishing period. Animals were housed in groups of four in 6 × 6 m pens with straw bedding, according to a balanced design relative to feeding treatments. The cows were housed in a freestall barn equipped with individual feed bunks and automatic gates. They were individually offered their appropriate allowance of concentrates and straw each day. Feed intake was calculated daily and adjusted periodically to ensure a daily gain of 1,150 g/d. The finishing period of 101 d, slightly longer than the French standards for cattle (approximately 70 d), was chosen to achieve good production conditions and to facilitate experimental organization.

Cows were finished and slaughtered under conditions of limited stress (n = 16) or physical and psychological stress (n = 16) as described in Bourguet et al. (2010). Groups included the same numbers of animals for each diet. For the limited stress condition, the cows were transported with a nonexperimental conspecific to avoid social isolation stress and were handled calmly. Specifically, for the stress condition, each cow was individually transported in a lorry (social isolation) towards an unfamiliar farm, where it was taken through a labyrinth by two purposefully noisy experimenters over a period of 30 min. The cow was then transported for 15 min to the experimental slaughterhouse. All cows were slaughtered without any electrical stimulation in compliance with INRAE ethical guidelines for animal welfare at the experimental slaughterhouse of INRAE. The cows were stunned by a captive bolt gun and exsanguinated, as performed in French commercial slaughterhouses. Carcasses were stored in a chilling room (4 °C) for approximately 45 min following exsanguination. Carcasses were sold for human consumption as in any controlled commercial slaughterhouse. Stress status was evaluated through the plasma and urinary cortisol and heart rate as described in Bourguet et al. (2010). Muscle samples from the M. semitendinosus (ST) (a himdmuscle involved in locomotion) and the M. longissimus thoracis (LT) (a support muscle for the body) were excised within 15 min after slaughter. They were immediately frozen in liquid nitrogen and stored at −80 °C until RNA extraction.

As indicated in Bourguet et al. (2010) and Delosière et al. (2020), experimental procedures and animal holding facilities respected French animal protection legislation, including licensing of experimenters. The procedures were controlled and approved by the French Veterinary Services (agreement B63 345 17). The animal experimental design was described in and registered in the research unit quality management database.

Muscle transcriptome analysis

Transcriptome analyses were carried out with Agilent oligonucleotide microarrays designed with 10,064 probes (including 1,614 control probes) for 4,210 bovine genes, including more than 3,000 specific muscular genes. The microarray was designed for monitoring transcriptional changes in genes involved in muscle growth (including energy and protein metabolism), carcass composition, fat metabolism and beef quality (including marbling). The microarray was first described in Hocquette et al. (2012a) and subsequently used in Costa et al. (2018).

RNA extraction, RNA quality checking and quantification, target amplification and labelling, hybridization with the probes, and extraction of fluorescent hybridization signals were performed as previously described in Hocquette et al. (2012a).

Data were preprocessed by Feature Extraction 10.1 software (Agilent Technologies, Santa Clara, CA, USA) for all samples and probes. The probes not meeting the quality criteria (saturation and uniformity of spots, intensity above background noise, etc.) were filtered out. Each array was normalized by dividing the raw intensity values of its probe by the median intensity of the control probes of the array. Each probe intensity was then normalized by dividing its raw value by the median of the corresponding probes from all arrays. After removing the probes with missing values, a log2 transformation was applied to the data.

The transcriptomic data were submitted to Gene Expression Omnibus (GEO) under the accession number GSE119912. Differential analyses were conducted via linear modelling with the diet supplementation* period * stress interaction factor to explain the probe levels. An empirical Bayes method was used to moderate the standard errors of the estimated log-fold changes using the R/Limma package (http://bioinf.wehi.edu.au/limma/) as described in Smyth, Yang & Speed (2003) with a Benjamini and Hochberg multiple testing correction (Benjamini & Hochberg, 1995). The genes for which at least 80% of the probes were consistently different at the adjusted p value 10% (i.e., estimated rate of true positives in the probe list of 90%) were retained and considered differentially expressed genes (DEGs). All probe ratios were found to be consistent for each DEG, meaning that for one gene, all probe ratios were lower than 1 or greater than 1.

Gene Ontology enrichment

Functional enrichment according to Gene Ontology Biological Process (GO BP) was assessed by submitting lists of accession numbers (for DEGs) or gene names (for common regulators and targets of the DEGs) to ProteINSIDEv2 (Kaspric et al., 2015, https://umrh-bioinfo.clermont.inrae.fr/ProteINSIDE_2/). This workflow enables the analysis of lists of protein or gene identifiers from ruminant species and gathers biological information provided by functional annotations, putative protein secretion and protein interactions. It queries the g:Profiler database based on the most complete information available for Bos taurus. The list of array probes was used as a background list for enrichment analysis of the DEG lists. The GO enrichment test was declared significant at a Benjamini–Hochberg FDR < 0.08 (i.e., estimated rate of true positives in the gene list of 92%). The results are expressed as –log10 (p value) on the graphs.

Identification of cis-transcriptional modules

Promoter sequences were extracted using the program Gene2Promoter (version 3.4.1; Genomatix Software Suite, Munich, Germany, www.genomatix.de) using the default settings, 500 bp upstream and 100 bp downstream of the transcription start site. We selected bovine promoters with at least one relevant transcript and preferentially a high quality level (experimentally verified 5′ transcript or with 5′ end confirmed by PromoterInspector prediction) and for whose number of conserved orthologous promoters was at least 50% of loci. This was performed for DEGs and for all of the genes in the microarray. As the coregulation of mammalian genes usually depends on a combination of TFs rather than individual TFs alone, cis-acting regulatory elements are often organized into frameworks of motifs called cis-transcriptional modules. The selected promoters were submitted to the ‘ModelInspector’ task of GEMS Launcher (version 4.1; Genomatix Software, Munich, Germany, www.genomatix.de) to search for cis-transcriptional modules. For this purpose, the promoter sequences of the genes were scanned for matches to the Promoter Module 5.4 Library (Vertebrate Module section). Fisher’s exact test was then used to identify overrepresented cis-transcriptional modules in the DEG set compared to the total gene set of the microarray.

Construction of interaction networks

Network analysis was performed with Pathway Studio software version 12.0.1.9 using Elsevier’s Resnet Mammal DataBase (Ariadne Genomics, Rockville, MD, USA). Gene interaction networks were built with the DEGs targeted by the most represented cis-transcriptional modules for each muscle (targeting at least 5 and 4 DEGs for LT and ST, respectively) and with the DEGs targeted by the nine overrepresented cis transcriptional modules common to both muscles, generating two muscle-specific networks and one common network of stress response. For each set of genes, to reconstruct the network, Pathway Studio searches known relations between the genes and adds regulators and the expression targets common to them. Filters were applied to identify only key expression regulators and targets of each network. To be added to the network, target genes must be linked to a minimum of three bibliographic references and have at least six known relations in the Pathway Studio Database. For regulators, three bibliographic references and two known relations are also needed, except for the regulators of the nine common modules for which a cut-off of five relations was chosen.

Finally, Venn diagrams were used to identify the major regulator genes and major target genes among the DEGs targeted by cis-transcriptional modules specific to LT and ST and in both muscles. Subnetworks between DEGs and their major regulator genes and between DEGs and their major target genes were extracted.

Validation of differential expression levels

A RT–qPCR assay was performed on the LT samples of 10 animals/group for 4 genes (ATF3, CEBPD, SMAD7 and FOS) with the StepOne Plus™ Real-Time PCR System using the Power SYBR1 Green master mix (both Applied Biosystems, Foster City, CA, USA). The GeNorm algorithm (Vandesompele et al., 2002) was used to determine the optimal number of reference genes required to effectively normalize the qPCR data. Four housekeeping genes were selected: UXT, MRPL39, CLN3 and TOP2B. The primer sequences (Table S1) with an annealing temperature of 60 °C were designed using Primer3 software. qPCR was performed using a StepOnePlus thermocycler (Applied Biosystems, Foster City, CA, USA). The PCR efficiency of each primer pair was tested with a 10-fold dilution series of purified cDNA. Each reaction was subjected to melting curve analysis to ensure the specificity and integrity of the PCR product. Student’s t test was used to test the significance of the difference between the limited stress and stress groups.

Identification of genes corresponding to DEGs in quantitative trait loci (QTLs)

A query of genetic information from the lists of the DEGs and the common regulators and targets of the DEGs was performed with the QTL module included in ProteINSIDEv2. Briefly, each DEG was searched on NCBI to retrieve the location of the corresponding gene on the genome. Then, this location was compared with the QTL positions in the QTL database “AnimalQTLdb”. The location on the genome must be included entirely in a QTL to consider that the DEG is mapped in the QTL.

Results

Transcriptomic profiles

We recorded changes in gene expression profiles in the LT and the ST. Individual data are available in the GEO repository under accession number GSE119912. No effect was detected for the EP diet, the stress*EP diet or the stress*diet in either muscle, but an effect of stress was detected (P < 0.1). In the stressed cows compared to cows handled with limited stress, microarray analysis revealed changes in the abundance of 67 transcripts in the LT (including 43 up- and 24 downregulated transcripts; P < 0.1, Table S2) corresponding to 62 DEGs with unique gene names (Fig. 1). In the ST, changes in 36 transcripts were detected (including 33 up- and 3 downregulated transcripts; P < 0.1, Table S2) corresponding to 32 DEGs with unique gene names (Fig. 1). Among the differentially expressed transcripts, 27 were common to both muscles, corresponding to 24 unique gene names (Fig. 1). They included eight known transcription factors (TFs): SMAD7, ETS2, MYOG, ATF3, HES6, CEBPD, HEYL, and FOS (Table S2). In addition, muscle-specific genes were differentially expressed according to the stress status (38 in the LT and 8 in the ST; Fig. 1). These genes included four TFs (MYOD1, MYF6, CEBPB, and HES1) and one transcription cofactor (MED23) in the LT and a transcriptional activator (ZNF750) in the ST. The differential abundance of four TF transcripts (ATF3, CEBPD, SMAD7 and FOS) was checked by qPCR experiments in the LT and confirmed the observed changes, as illustrated in Table 1.

Figure 1 Venn diagram visualizing the intersection of the lists of the gene names of the differentially expressed genes (DEGs) in response to pre slaughter stress in the M. longissimus thoracis (LT) and in the M. semitendinosus (ST).

A subset of 24 common DEGs was assigned to a set of core stress responsive genes. The two subsets of DEGs only in the LT (n = 38) or in the ST (n = 8) were considered as components of the muscle-specific response to stress. Transcriptional regulators are underlined: Transcription factor (unbroken line), transcriptional modulator (dotted line).

Table 1 Validation of some differentially expressed genes following pre-slaughter stress in the Longissimus thoracis muscle.

The abundance of some DEGs detected by microarray analysis was quantified by qRT PCR in the Longissimus thoracis muscle of stress cows vs. cows handled with limited stress (2n = 20). Variation of reference genes used for normalization was computed with the GeNorm software package. Student t-test was used to test the significance of the difference between the two conditions.

	Fold change (qPCR)	P-value	Fold change (Microarray)	
ATF3	2.1	0.006	2.6	
CEBPD	4.	0.001	3.6	
FOS	1.4	0.143	2.5	
SMAD7	1.8	0.006	1.7	

Lists of DEGs according to stress status were submitted to biological information mining through Gene Ontology (GO) term enrichment compared to the background list of the microarray (Data S1). In LT, 9 GO biological process (GO BP) terms were enriched (P < 0.08). In the ST, 26 GO BP terms were enriched (P < 0.08). As illustrated in Fig. 2, 9 of these GO terms identified in both the LT and ST: regulation of gene expression (23 genes in the LT, 13 in the ST), transcription by RNA polymerase II (16 genes in the LT, 12 in the ST), regulation of transcription by RNA polymerase II (16 genes in the LT, 12 in the ST), regulation of biosynthetic process (21 genes in the LT, 14 in the ST), regulation of cellular biosynthetic process (20 genes in the LT, 13 in the ST), regulation of macromolecule biosynthetic process (20 genes in the LT, 13 in the ST), regulation of cellular macromolecule biosynthetic process (20 genes in the LT, 13 in the ST), skeletal muscle cell differentiation (4 genes in the LT, 3 in the ST), and muscle organ development (8 genes in the LT, 5 in the ST). For each considered GO BP, the list of genes included both common and muscle-specific DEGs. There were no significant GO cellular component terms at P < 0.08 in both the LT and ST muscles (data not shown). In the LT, 18 GO molecular factor (MF) terms were enriched (P < 0.08). In the ST, 19 GO MF terms were enriched (P < 0.08). Sixteen of these GO terms were enriched in both the LT and ST. They mainly refer to DNA-binding and transcription regulator activity (Supplemental Dataset).

Figure 2 Common GO terms across muscles for the differentially expressed genes (DEGs) in response to preslaughter stress.

Lists of DEGs were submitted to functional annotation compared to the microarray background (data available in Additional File 3). The intersection of the lists of GO terms was computed at http://bioinformatics.psb.ugent.be/webtools/Venn/. Gene names capitalized in bold are common DEG between muscles. LT: M. longissimus thoracis; ST: M. semitendinosus.

Cis-transcriptional modules

A promoter analysis was performed with Gene2promoter of the Genomatix Software Suite to identify common TF binding sites in the promoter regions of genes–called cis transcriptional modules–that may account for coregulation among differential transcripts. For 52 of the DEGs in the LT, 168 promoters were retrieved from the Genomatix Promoter Database, of which 111 were selected for further analysis (Table S3). ModelInspector enabled the retrieval of 288 different cis-transcriptional modules (at 1,378 locations). For 28 of the DEGs in the ST, 84 promoters were retrieved. Of these, 57 promoters were further analysed with ModelInspector, and 201 cis-transcriptional modules were found (on 675 match positions). The same analysis was performed for all the genes represented on the microarray. As illustrated in Table S3, 24 cis-transcriptional modules were identified as overrepresented DEGs compared to the genes represented on the microarray (P < 0.1) in the LT and 25 in the ST. Nine of the modules were overrepresented in both muscles. The cis-transcriptional modules and the DEGs targeted by these modules in each muscle as identified by ModelInspector are listed in Table 2. Cis-transcriptional modules with binding sites for TFs in the ETS family and SP1 family had a high occurrence in the promoters of the DEGs in both muscles.

Table 2 Over-represented transcriptional modules in the promoter of the stress-responsive genes in the muscles of cows.

The transcriptional modules were searched with the module inspector function of Genomatix, their occurrence was examined in the promoters of genes of the experimental datasets and the number of target genes was determined in each dataset.

Muscle	Module	p value	Occurrence
of module	Number of target genes	Gene ID	
LT	ETSF_ETSF_01	0.076	16	15	IL16 SERPINE1 HES1 HSPBAP1 CDIPT XYLT2 TUBB3 THBS1 MED23 GPAM PIGM CEBPD HEYL HES6 MYOD1	
SP1F_CAAT_02	0.040	10	9	NME6 SDC4 PDPR HES6 PFKFB3 CDIPT PMP22 THBS1 IFRD1	
CAAT_AP1F_01	0.035	8	8	SLC25A25 SERPINE1 NME6 IMP3 HSPBAP1 SLC2A3 THBS1 ATF3	
SP1F_EBOX_SP1F_01	0.024	8	7	DFFB GLUL PDK4 IMP3 PMP22 XYLT2 CEBPD	
CAAT_SP1F_01	0.088	5	5	SERPINE1 ATP1B1 GLUL GEM HES6	
GATA_GATA_GATA_01	0.037	5	3	NME6 SLC16A6 MED23	
YY1F_SRFF_02	0.016	3	3	SLC2A3 ATF3 FOS	
SORY_SORY_EGRF_01	0.061	3	3	MUSK ATP1B1 RAB3IL1	
NFKB_NFKB_01	0.064	3	3	SLC25A25 GLUL GEM	
HNF1_GATA_01	0.098	3	3	MED23 PLD1 ATP1B1	
KLFS_NR2F_KLFS_01	0.024	3	2	SERPINE1 TUBB3	
STAF_SP1F_01	0.026	2	2	GLUL HEYL	
RXRF_EBOX_01	0.043	2	2	PDPR RAB3IL1	
AP1F_SMAD_01	0.055	2	2	IL16 THBS1	
ETSF_AP1F_04	0.067	2	2	ACOT11 HSPBAP1	
CEBP_MYBL_03	0.076	2	2	ACOT11 HSPBAP1	
AARF_CEBP_01	0.091	2	2	ABRA NME6	
BRNF_RXRF_02	0.066	4	1	DLL4	
NFKB_ETSF_01	0.007	2	1	DLL4	
SRFF_AP1F_01	0.047	1	1	FOS	
ETSF_SP1F_SMAD_01	0.062	1	1	HEYL	
YY1F_SRFF_01	0.076	1	1	FOS	
PAX8_NKXH_01	0.076	1	1	PMP22	
ETSF_SRFF_01	0.091	1	1	FOS	
ST	SP1F_SP1F_06	0.002	30	14	PGF GADD45A SLC16A6 ADAMTS9 CYP1A1 SLC2A8 SDC4 PMP22 TUBB3 IFRD1 HYAL2 ATF3 HES6 HEYL	
NFKB_SP1F_03	0.002	12	8	SLC2A8 SDC4 MYLK4 PGF LRP4 PMP22 HEYL CEBPD	
SP1F_ETSF_04	0.087	8	8	ABRA SDC4 PGF LCAT PMP22 CYP1A1 SMAD7 HES6	
SMAD_E2FF_01	0.088	12	7	SLC2A8 SDC4 IFRD1 PDK4 CEBPD HES6 FOS	
SP1F_YY1F_01	0.044	10	7	PGF ABRA GEM SDC4 SLC2A8 ATF3 HES6	
SP1F_CAAT_02	0.039	6	5	SDC4 PFKFB3 IFRD1 PMP22 HES6	
SP1F_EBOX_SP1F_01	0.085	4	4	MYLK4 PMP22 PDK4 CEBPD	
RUSH_EGRF_01	0.049	3	3	SDC4 GADD45A SPOCK2	
IRFF_NFAT_01	0.084	3	3	MYLK4 ADAMTS9 IFRD1	
GATA_GATA_GATA_01	0.013	4	2	SLC16A6 ADAMTS9	
	MYOD_MYOD_03	0.066	3	2	SPOCK2 HES6	
AP1F_ETSF_04	0.013	2	2	IFRD1 HYAL2	
YY1F_SRFF_02	0.028	2	2	ATF3 FOS	
ZFHX_ZFHX_NKXH_01	0.037	2	2	GADD45A ADAMTS9	
	SMAD_HIFF_01	0.032	2	1	PFKFB3	
SP1F_MZF1_01	0.035	2	1	PMP22	
ETSF_SP1F_SMAD_01	0.016	1	1	HEYL	
SRFF_AP1F_01	0.024	1	1	FOS	
YY1F_SRFF_01	0.039	1	1	FOS	
PAX8_NKXH_01	0.039	1	1	PMP22	
ETSF_SRFF_01	0.047	1	1	FOS	
MEF2_MYOD_01	0.054	1	1	SLC16A6	
KLFS_CREB_KLFS_01	0.070	1	1	SLC2A8	
CAAT_SREB_01	0.077	1	1	IFRD1	
GATA_HNF1_02	0.077	1	1	PDK4	
Notes:

LT: Longissimus thoracis muscle; ST: Semitendinosus muscle.

Modules in bold were in common between muscles.

TFs genes are underlined.

Interaction networks and identification of regulators and main targets of DEGs

Finally, with Pathway Studio 2, we constructed interaction networks between the DEGs targeted by the overrepresented cis-transcriptional modules for each muscle and between the DEGs targeted by the 9 overrepresented cis-transcriptional modules common to both muscles. We thus generated muscle-specific networks and one core network of stress responses. Then, using the Pathway Studio 2 database, we searched for the main regulators and the main targets of the 3 networks (Data S2, sheets 1–6). We next identified the similarities among the lists obtained from these datasets to identify the key common regulators and targets (Data S2, sheets 7–8). Ten main regulators of stress-responsive genes were identified: AKT1, EGF, HIF1A, IFNG, IL1B, INS, MAPK1, MAPK14, TGFB1, and TNF. GO mining showed that they were highly related (P < 0.001) to the regulation of oxidoreductase activity and specifically the regulation of monooxygenase activity, inflammatory response, immune system, and carbohydrate metabolism (transport and metabolic processes) (Data S2, sheet 7). A list of 14 main targets of the DEGs was identified: BCL2, BGLAP, CDKN1A, COL3A1, ERBB2, FN1, ICAM1, IL6, MMP2, PPARG, SELE, SLC2A4, TLR4, and VEGFA. GO mining showed that they were highly related (P < 0.001) to response to cytokine, response to oxygen levels, response to glucocorticoids, response to stress, immune system, angiogenesis, and carbohydrate homeostasis (Data S2, sheet 8). The networks of the DEGs and the common regulators and of the DEGs and common targets are presented in Fig. 3. The list of DEGs (LT-specific, ST-specific and common DEGs), the main regulators and the main targets of the DEGs are summarized in Table 3. Twenty-four of the corresponding genes were located in QTLs linked to meat quality: shear force (CDIPT, CEBPD, DNAJB4, GPAM, RAB3IL1, MAPK1, and TLR4), muscle compression (ADRB2), tenderness score (ADRB2, CDIPT, RAB3IL1, and IFNG), muscle pH (DLL4, ERBB2), juiciness (ATP1B1, DFFB, RAB3IL1, SELE), and marbling (CEBPD, DLL4, ERBB2, GADD45, ICAM1, IL1B, IL6, LEAP2, MYF6, PDK4, PMP22, SMAD7, TNF). Some genes were also mapped in QTLs associated with marbling score (CEBPD, DLL4, DNAJB4, GADD45A, LEAP2, MYF6, PDK4, PIGM, PMP22, RAB3IL1, and SMAD7) and with lipid class contents (e.g., monounsaturated fatty acid content and conjugated linoleic acid content: ADRB2; omega-6 to omega-3 fatty acid ratio and palmitic acid content: LCAT; trans-11, cis-15-C8:2 fatty acid content: GADD45A).

Figure 3 Common regulators and common targets between the DEGs targeted by cis-modules in the LT and in the ST.

(A) Common regulators between the DEGs targeted by the most represented cis modules for the LT (highlighted in blue) and the ST (highlighted in green), and the DEGs targeted by the cis-modules common to both muscles (highlighted in yellow). The list of 10 potential regulators of stress responsive genes included: AKT1, EGF, HIF1A, IFNG, IL1B, INS, MAPK1, MAPK14, TGFB1, and TNF. (B) Common targets between the DEGs targeted by the most represented cis modules for the LT (highlighted in blue) and the ST (highlighted in green), and the DEGs targeted by the cis modules common to both muscles (highlighted in yellow). The list of 14 potential targets of the DEGs included: BCL2, BGLAP, CDKN1A, COL3A1, ERBB2, FN1, ICAM1, IL6, MMP2, PPARG, SELE, SLC2A4, TLR4, and VEGFA.

Table 3 Components of the molecular response initiated by pre slaughter stress in two muscles of cows as revealed by transcriptomic signatures.

The differentially expressed genes (DEGs), the main regulators and the main targets of the 3 datasets of the DEGs as identified by Pathway Studio are listed. Query of genetic information was performed with the QTL module included in ProteINSIDE in order to retrieve information on the location of the genes encoding proteins of interest within published Quantitative trait loci (QTLs) for meat and carcass. This module interrogates the publicly available QTL database “Animal QTLdb”.

Type of response	Gene name	Transcription regulator	Location in a bovine QTL	
LT specific DEG	ACOT11			
ADRB2		Tenderness score, Muscle compression, Saturated fatty acid content, Conjugated linoleic acid content	
ARL6IP2			
ATP1B1		Juiciness	
CDIPT		Tenderness score, Shear force	
CEBPB	TF		
CXCR6			
DFFB		Juiciness	
DLL4		Muscle pH, Marbling score	
DNAJB4		Shear force	
GLUL			
GPAM		Shear force	
HES1	TM		
HSPB1*			
IDS			
IL16			
IMP3			
ITGAE			
LEAP2		Marbling score	
MED23	TM		
MUSK			
MYF6		Marbling score	
MYLC2			
MYOD1	TF		
NME6			
NOL6			
PDPR			
PIGM			
PITPNM2			
PLD1			
	RAB3IL1		Tenderness score, Shear force, Juiciness, Marbling score	
SERPINE1			
SLC25A25			
SLC2A3			
THBS1			
	TREM1			
XYLT2			
YWHAZ			
ST-specific DEG	ADAMTS9			
ATL2			
CYP1A1			
HYAL2			
MYLK4			
PPP2B			
SLC2A8			
ZNF750	TM		
Common DEG	ABRA			
ATF3	TF		
CEBPD	TF	Shear force, Marbling score	
ETS2	TF		
FOS	TF		
GADD45A		Marbling score	
GEM			
HES6	TF		
HEYL	TF		
IFRD1			
LCAT			
LRP4			
MYOG	TF		
PDK4		Marbling score	
PFKFB3			
PGF			
PMP22		Marbling score	
RGS2			
SDC4			
SLC16A6			
SMAD7	TF	Marbling score	
SORBS1			
SPOCK2			
TUBB6			
Common main regulators	AKT1			
EGF			
HIF1A	TF		
IFNG		Tenderness score	
IL1B		Marbling score	
	INS			
MAPK1		Shear force	
MAPK14			
TGFB1			
TNF		Marbling score	
Common main targets	BCL2			
BGLAP			
CDKN1A			
COL3A1			
ERBB2		Muscle pH, Marbling score	
FN1			
ICAM1		Marbling score	
IL6		Marbling score	
MMP2			
PPARG	TF		
SELE		Juiciness	
SLC2A4			
TLR4		Shear force	
VEGFA			
Notes:

Only QTL related to meat quality are shown in the table.

* Proposed as a protein biomarker for high ultimate pH (pHu) meat in Sentandreu et al. (2021).

Discussion

Transcriptional response to stress

Understanding how preslaughter stress impacts muscle physiology would provide information for the management of beef quality, especially tenderness. In this study, we compared the muscle transcriptional profiles of cows exposed to preslaughter emotional and physical stress with those of control cows handled with limited stress. We hypothesized that this approach may be useful for investigating the molecular mechanisms of the stress response and the potential impact on meat quality. We identified changes in the abundance of several gene transcripts in two muscles of the cows exposed to stress. We found evidence of a common transcriptional response in both muscles, albeit with different metabolic types and activities, even though some muscle-specific DEGs were detected. Notably, there was a core stress response in both muscles, as shown by common DEGs and common GO terms (mainly related to the regulation of gene expression and muscle development). The highest number of DEGs was detected in the LT. This may be related to the high oxidative metabolism of the LT (Hocquette et al., 2012b), which makes it more prone to changes in oxidative status and therefore susceptible to stress. Muscle gene expression in response to stress likely also depends on sex (Oster et al., 2014), the nature and intensity or duration of the stress, and breed, which may explain some differences in the results of our study and a previous study on Angus animals (Zhao et al., 2012). Nevertheless, the regulation of genes involved in carbohydrate, lipid, and protein metabolism is likely to occur in many cases, as observed in this study, as well as in other studies on cattle (Buckham Sporer et al., 2007; Zhao et al., 2012) and pigs (Davoli et al., 2009).

Newly translated transcription factors and their related biological pathways

While the short-term response to stress may be primarily driven by changes in protein phosphorylation (e.g., reversible phosphorylation (Mato et al., 2019)), as well as enzyme activity or protein abundance, our study provided convincing evidence that the response to stress includes a transcriptional component, as previously reported in two studies (Davoli et al., 2009; Zhao et al., 2012). Indeed, functional annotation of the lists of DEGs revealed the enrichment of GO terms related to the regulation of gene expression and transcription. It is well accepted that the primary response to stress involves the activation of pre-existing TFs by phosphorylation (Sabban & Kvetňanský, 2001). Our data indicate that newly translated TFs may also relay the stress response, as 11 of the DEGs encode TFs. Eight of these TFs were identified in both muscles, of which some were detected as nodes in the molecular networks associated with the response to stress. The majority of the differentially expressed TFs were upregulated, except for two muscle regulatory factors (MYOG and MYOD1) and a transcriptional repressor (HES1). MYOG and MYOD1 are basic helix-loop-helix family TFs essential for myogenesis, including during the regenerative process (Zammit, 2017). HES1 is a downstream target of Notch (Borggrefe & Oswald, 2009). It is also a master regulator of glucocorticoid receptor-dependent gene expression. It is silenced by the primary stress hormones glucocorticoids (Revollo et al., 2013). The observed downregulation of HES1 was not surprising since stressed cows showed high plasma and urinary cortisol levels (Bourguet et al., 2010). Of the upregulated TFs, 4 were basic leucine zipper (bZip) TFs: FOS, ATF3, CEBPB, and CEBPD. ATF3, a member of the mammalian cAMP-responsive element-binding protein (CREB) family, is induced by various stresses. ATF3 is a sensor for a wide range of conditions and modulates the immune response, atherogenesis, cell cycle, apoptosis, and glucose homeostasis (Jadhav & Zhang, 2017). ATF3 has been considered an adaptive response gene with a dual mode of action to activate (as a homodimer) or repress (as a heterodimer) target gene expression. It was proposed that ATF3 functions as a “hub” of the cellular adaptive-response network that helps cells adapt to disturbances of homeostasis (Hai, Wolford & Chang, 2010). ATF3 was also found to be differentially expressed following acute stress induced by surgery in Angus beef (Zhao et al., 2012). The bZip proteins CEBPB and CEBPD are members of the C/EBP family, which participates in a number of biological responses, including energy metabolism, cell proliferation and differentiation, and immune responses (Ramji & Foka, 2002). Their binding sites are found in the regulatory regions of a large number of acute phase proteins. A dual role was proposed for the C/EBP proteins as mediators of both inflammatory responses and glucocorticoid effects (Nerlov, 2007; Roos & Nord, 2011). CEBPD expression is induced by inflammatory effectors and hypoxia and promotes proinflammatory signalling and hypoxia adaptation (Balamurugan & Sterneck, 2013). CEBPB was also recently identified as a novel regulator of satellite cell homeostasis that promotes differentiation at the expense of self-renewal (Lala-Tabbert et al., 2016).

Overrepresented TF binding sites in the promoters of DEGs and related biological pathways

Several cis-transcriptional modules were located in the promoters of the DEGs. Modules common to both muscles were detected mainly in the promoters of common DEGs, while muscle-specific cis-transcriptional modules were detected in the promoters of muscle-specific DEGs. However, some specific cis-transcriptional modules were detected in the promoters of common DEGs. FOS was targeted by 4 common modules in both muscles plus 1 specific module in the ST. ATF3 was targeted by 1 common module in both muscles and by 3 specific modules (1 in the LT and 2 in the ST). Examination of cis-transcriptional modules of DEGs from both muscles revealed that binding sites for the transcription factor SP1 and for members of the ETS family are often included in those modules. SP1 is ubiquitously expressed and, in addition to functioning as a ‘housekeeping’ TF, may be a key mediator of gene expression induced by insulin and other hormones (Solomon et al., 2008). ETS1 is a highly conserved TF that controls the expression of cytokines, chemokines and angiogenesis factors (Russell & Garrett-Sinha, 2010). ETS binding sites were found in the promoter of common differentially expressed TFs as well as in the promoter of 11 of the 15 LT-specific DEGs.

Other biological pathways related to the response to stress

Our study provided additional evidence that the response to stress interacts with the immune response, inflammatory response, and chemotaxis, as well as the production of interleukins (IL-16 in the LT; IL-1 B and IL-6 as main regulators and targets of the DEGs; and IL-10 and IL-13 in the list of common targets of the DEGs). This is consistent with previous studies examining the response to stress in livestock animals: A transcriptional shift in acquired and innate immunity pathways was reported in the peripheral blood of psychosocially stressed pigs (Oster et al., 2014). Amplified inflammatory activity was also detected in blood neutrophil expression in young bulls following truck transportation for 9 h (Buckham Sporer et al., 2007) and in the LT muscle of Angus beef exposed to acute stress induced by surgery (Zhao et al., 2012). Moreover, a conserved transcriptional response to chronic social stress involving increased expression of proinflammatory genes (including IL-6 and IL-8) has been reported in blood leukocytes (Powell et al., 2013) in mice and humans. In our study, changes in chemokine and cytokine expression in muscle were most likely part of the adaptive mechanisms contributing to the stress response (Fig. 3). IL-16 is a lymphocyte chemoattractant factor also classified as an “alarmin” (Rider et al., 2017). IL-6 and IL-8 are also regarded as myokines released from muscle in response to contractions (Brandt & Pedersen, 2010). Muscle-derived IL-6 may mediate some of the anti-inflammatory and insulin-sensitizing effects of physical exercise (Covarrubias Anthony & Horng, 2014).

Another striking result of our study is the upregulation of transcripts related to the carbohydrate metabolic pathway, e.g., transcripts encoding PDK4 (an inactivator of pyruvate dehydrogenase complex; targeted by 1 cis-transcriptional module in the LT and 3 modules in the ST), PFKFB3 (a glycolysis regulator; targeted by 1 cis-transcriptional module in the LT and 2 cis-transcriptional modules in the ST) and SLC25A25 (a mitochondrial ATP transporter; targeted by 2 cis-transcriptional modules in the LT). This illustrates a switch in energy metabolism in the muscles of animals exposed to exercise and psychological stress towards anaerobic metabolism to support ATP production for muscle contraction. PDK4 plays a pivotal role in controlling metabolic flexibility (Zhang et al., 2014), and its expression is increased in response to moderate intensity exercise.

Analysis of molecular networks also highlighted the contribution of the response to oxygen levels/hypoxia in the response to stress, albeit via different transcripts and different contractile and metabolic muscle types. Consistently, the TF hypoxia inducible factor (HIF1A) was identified as a main common regulator of the DEGs. This could be a signature of oxygen imbalance or of the physical activity imposed on the cows. Thus, it may not be surprising that the expression of PFKB3, a downstream target of HIF, was upregulated. Hypoxia was also demonstrated to cross-talk with the Notch signalling pathways, which regulate satellite cell quiescence and self-renewal (Liu et al., 2012). Since quiescent satellite cells have a low metabolic rate, fewer mitochondria and anaerobic metabolism, this is likely part of the adaptive signature of muscle to stress. Thus, the combined signatures of hypoxia, the Notch signalling pathway (Fukada et al., 2007), MYOD1 downregulation (Kopan, Nye & Weintraub, 1994), and IRFD1 (an inducer of regenerative myogenesis) downregulation further indicate that quiescent satellite cells are stress targets and most likely physical activity targets.

Putative effects on meat quality

Finally, the transcriptomic muscle response to preslaughter stress may have an impact on meat quality through energy metabolism and hypoxia. Indeed, anaerobic glycolysis is highly relevant to beef quality since it is involved in postmortem protein degradation and hence beef tenderization during meat ageing (reviewed by Maltin et al., 2003). This process is regulated by the decline in muscle pH due to the conversion of glycogen into lactate following the lack of oxygen after slaughtering. Stress was shown to markedly affect meat tenderness by increasing postmortem ultimate pH (Purchas, 1990) due to the depletion of glycogen stores by stress prior to slaughtering, which leads to dark-cutting meats. Reliable indicators of the occurrence of high pH and preslaughter stress were identified in the sarcoplasmic proteome of muscle (Fuente-Garcia et al., 2019; Sentandreu et al., 2021). They were mainly involved in metabolism, chaperone- and stress-related processes, muscle contractility/fibre organization, and transport activities. In our study, several genes encoded by the DEGs and the common regulators or targets of the DEGs were located in bovine QTLs of the meat and carcass group associated with meat quality parameters known to be impacted by stress: muscle pH (DLL4, ERBB2), shear force (CDIPT, CEBPD, DNAJB4, GPAM, RAB3IL1, MAPK1, and TLR4), tenderness score (ADRB2 CDIPT RAB3IL1, and IFNG), compression (ADRB2), and juiciness (ATP1B1, DFFB, and RAB3IL1). Some genes were also mapped in QTLs associated with marbling score and with lipid class contents. However, the relationships between transcript levels and these meat quality parameters remain to be studied.

Conclusions

Exposure to emotional stress (novelty, social isolation, presence of active humans, noise) and physical effort (walking) prior to slaughter induced a transcriptional response in two muscles in cows. Our data provide evidence of a coordinated response in two muscles of stressed animals due to the identification of common target genes, associated functions, cis-transcriptional modules, regulators and downstream targets. The response included an interplay between metabolic changes (glycolytic), hypoxia, inflammatory process, and satellite cell renewal/quiescence, likely due to elevated cortisol. However, the relative contribution of mechanisms related to stress and to physical activity induced by walking the labyrinth remains to be elucidated.

From an animal production perspective, the identification of gene networks activated by stress will improve the understanding of the molecular mechanisms of meat conversion and beef quality defects caused by preslaughter stressful conditions suffered by cattle. The target stress-responsive gene network could be modulated by management factors (on farm nutrition, antioxidant supplementation, etc.) to reduce the adverse impact of stress.

Supplemental Information

Supplemental Information 1 Gene Ontology (GO) term enrichment of the DEG compared to the background list of the microarray.

Click here for additional data file.

Supplemental Information 2 Interaction networks and identification of regulators and mains targets of DEG.

Click here for additional data file.

Supplemental Information 3 Sequence of the primers used in qPCR experiments.

The sequence of the primers was designed using the Primer 3 software.

Click here for additional data file.

Supplemental Information 4 Differential genes in response to stress in the Longisssimus thoracis (LT) muscle and Semitendinosus (ST) muscle in stressed cows compared to controls.

Differential expression was computed on the GSE119912 using the LIMMA method with a Benjamini–Hochberg (BH) multiple testing correction. Genes for which 80% were differential probes at the adjusted p-value BH<10% were retained.

Click here for additional data file.

Supplemental Information 5 Over-represented transcriptional cis-modules in the dataset of stress responsive genes in the LT and in the ST muscles.

Promoters of the differentially expressed genes (DEG) were examined for transcriptional cis- modules using the Genomatix software suite. Detection of over-represented modules in the DEG was made by comparison to the total genes of the microarray.

Click here for additional data file.

Supplemental Information 6 qPCR raw data.

Click here for additional data file.

Supplemental Information 7 MIAME checklist.

Click here for additional data file.

Supplemental Information 8 Author checklist.

Click here for additional data file.

Supplemental Information 9 GO_MF_Data.

Enrichment in GO Molecular Factor terms in the lists of DEGs in both the LT and the ST muscles

Click here for additional data file.

The authors thank Geneviève Gentes and the HELIXIO Company (formerly Imaxio, Clermont-Limagne Biopôle) for performing the transcriptomic analyses and Dominique Bauchart, Denis Durand, Claudia Terlouw, Brigitte Picard and Joelle Henry-Berger for scientific discussions.

Additional Information and Declarations

Competing Interests

Author Contributions

Animal Ethics

Data Availability

The authors declare that they have no competing interests.

Isabelle Cassar-Malek conceived and designed the experiments, performed the experiments, analyzed the data, prepared figures and/or tables, authored or reviewed drafts of the paper, and approved the final draft.

Lise Pomiès analyzed the data, prepared figures and/or tables, authored or reviewed drafts of the paper, and approved the final draft.

Anne de la Foye analyzed the data, authored or reviewed drafts of the paper, and approved the final draft.

Jérémy Tournayre analyzed the data, authored or reviewed drafts of the paper, and approved the final draft.

Céline Boby performed the experiments, analyzed the data, prepared figures and/or tables, authored or reviewed drafts of the paper, and approved the final draft.

Jean-François Hocquette conceived and designed the experiments, performed the experiments, analyzed the data, authored or reviewed drafts of the paper, and approved the final draft.

The following information was supplied relating to ethical approvals (i.e., approving body and any reference numbers):

The experiment was conducted in 2007. Before 2013 submission to the Ethics Committee was not mandatory in France. However, the Animal Facility of INRAE is approved for animal experimentation and that the scientists and technicians were authorized for animal experimentation. Experimental procedures and animal holding facilities respected French animal protection legislation, including licensing of experimenters. French Veterinary Services controlled and approved them (B63 345 17).

The following information was supplied regarding data availability:

The raw data are available in the Supplemental File. The transcriptomic datasets are available at GEO: GSE119912.

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
