# Peer review of "Transcriptome profiling reveals stress-responsive gene networks in cattle muscles"

_PeerJ, doi:10.7717/peerj.13150_

## Round 0.1 · original submission · Minor Revisions

Two reviewers have reviewed your manuscript. As both of the reviewers consider your manuscript has the merit to be published, however, they also indicated some weaknesses of the manuscript.

Reviewer 1 ·

Basic reporting

The writing of manuscript needs to be improved throughout. Sentences should be clearer, and more concise. Several suggestions and comments need to be addressed.
1. The abstract is recommended to be rewritten and should be clearer, and more concise.
2. The introduction should be supplemented with the latest references, such as lines 43-48, line 50, lines 53-58, lines 59-64.
3. In animals and samples section, the language is too cumbersome and should be concise and clear.
4. For all Table and Figures, further optimization is required in terms of resolution, layout and fonts.
5. Line 246, line 248, Lines 263-264, the p-value is 0.1 or 0.08, is there any reference basis? What was the impact on the results? Is the significance statistically significant?
6. Line 226, line 302, line 307, etc. The gene symbol should be Italic across manuscript, and there are many similar formatting problems, please check them.

Experimental design

The purpose of the study, the experimental design and the results all make sense. However, in muscle transcriptome analysis, why the authors did not use RNA-Seq, the choice bovine microarray strategy, there are many limitations. Due to the transcription of genetic information has space-time specificity and tissue specificity, a lot of gene information will be lost.

Validity of the findings

no comment

Additional comments

no comment

Reviewer 2 ·

Basic reporting

Please have a second set of eyes reading your manuscript to avoid grammar errors.

Experimental design

The manuscript titled “Transcriptome profiling reveals stress responsive gene networks in cattle muscles” presents results from a microarray analysis where transcriptome data from Longissimus thoracis muscle (LT) and the Semitendinosus muscle (ST) of cows exposed to stress (n=16) or not (n=16) preslaughter. The study connects gene expression data with QTL for meat tenderness and analyzes significant TFs. I would like to see more details on how the connections between DEGs and QTLs were established. Why only QTLs for meat tenderness were considered? There are many other QTLs affected by stress pre-slaughter that could be considered to make this analysis more complete.

Validity of the findings

The cutoff used to determine DEGs is confusing o me. It is mentioned that a P < 0.1 was considered (Line 246), I would like to know if a cutoff for FDR was also considered to establish the number of DEG for each comparison analyzed (EP diet, stress*EP diet, stress*diet or stress). In line 180 is written that “GO enrichment test was declared significant for P value Benjamini-Hochberg FDR < 0.05 or 0.08”. I would like to know if this FDR cutoff was also considered to calculate the number of DEG used in the GO enrichment analysis. It is not clear to me in the way it is written.

The decision of applying dietary treatments with vitamin E and polyphenols could be discussed in more detail. What was your expectation/hypothesis by supplementing them? References?

Functional enrichment using Wikipathways: I will recommend not to use Wikipathways; instead, consider using not only biological process GO term but also molecular functions and cellular component GO terms. It is common to see these 3 GO terms analyzed together. Even though it is an R application, Wikipathways decreases the quality of the manuscript because WikiPathways is a database of biological pathways maintained by and for the scientific community. Therefore, there are many hands changing these pathways constantly and makes me hard to fully trust in reliability of these pathways and that they were utilized in the right context.

Additional comments

Electronic feeding gates, please specify.

---

## Round 0.2 · accepted · Accept

Congratulations to all the authors that your manuscript has been accepted for publication.